# Towards Migrating Neural Network Implementations

Nadia Daoudi
nadia.daoudi@list.lu
Luxembourg Institute of Science and
Technology
Luxembourg

Iván Alfonso
ivan.alfonso@list.lu
Luxembourg Institute of Science and
Technology
Luxembourg

Jordi Cabot
jordi.cabot@list.lu
Luxembourg Institute of Science and
Technology
Luxembourg
University of Luxembourg
Luxembourg

## Abstract

The development of smart systems (i.e., systems enhanced with AI components) has thrived thanks to the rapid advancements in neural networks (NNs). A wide range of libraries and frameworks have consequently emerged to support NN design and implementation. The choice depends on factors such as available functionalities, ease of use, documentation and community support. After adopting a given NN framework, organizations might later choose to switch to another if performance declines, requirements evolve, or new features are introduced. Unfortunately, migrating NN implementations across libraries is challenging due to the lack of migration approaches specifically tailored for NNs. This leads to increased time and effort to modernize NNs, as manual updates are necessary to avoid relying on outdated implementations and ensure compatibility with new features. In this paper, we propose an approach to automatically migrate NN code across deep learning frameworks. Our method makes use of a pivot NN model to create an abstraction of the NN prior to migration. We validate our approach using two popular NN frameworks, namely PyTorch and TensorFlow. We also discuss the challenges of migrating code between the two frameworks and how they were approached in our method. Experimental evaluation on five NNs shows that our approach successfully migrates their code and produces NNs that are functionally equivalent to the originals. Our artefacts are available online.

## CCS Concepts

• **Computing methodologies** → **Neural networks**; • **Software and its engineering** → **Model-driven software engineering**; **Maintaining software**.

## Keywords

Migration, Neural Networks, MDE, Model Transformation, Code Generation

**ACM Reference Format:**
Nadia Daoudi, Iván Alfonso, and Jordi Cabot. 2026. Towards Migrating Neural Network Implementations. In *Proceedings of the 3rd ACM International Conference on AI-Powered Software (AIware '26), July 6–7, 2026, Montreal, QC, Canada.* ACM, New York, NY, USA, 10 pages. https://doi.org/10.1145/3805760.3814917

## 1 Introduction

With the growing demand for smart software, organizations are investing in developing and maintaining AI-based applications to respond to evolving customer needs and remain competitive in the market [5, 8]. This increasing demand is driven by the need to leverage the vast and ever-growing amounts of data to provide personalized services and enable more informed decision-making [45, 55]. Moreover, efficiency improvements and cost reduction are also considered significant drivers for adopting smart systems as they allow organizations to increase productivity using fewer resources [18, 59].

Neural networks are at the core of many smart systems, forming the foundation of their decision-making processes. They have been applied to various domains, consistently achieving state-of-the-art performance in tasks like image classification, speech recognition, and anomaly detection. Various types of NN layers exist, each designed for a specific task, enabling NNs to efficiently tackle a wide range of challenges. For instance, convolutional neural networks (CNNs) [31] are composed of convolutional and pooling layers that efficiently process and analyse spatial data, making them particularly effective for tasks like object detection [20] and image segmentation [39]. Recurrent neural networks (RNNs) [54], on the other hand, contain recurrent layers that excel in processing temporal and sequential data, which makes them suitable for tasks like machine translation [4] and time series prediction [27].

Neural network development is supported by a range of deep learning (DL) frameworks specifically designed for NN creation, training and validation, making it easier to implement and leverage advanced concepts and features effectively. With the rapid pace of artificial intelligence (AI) development, organizations may seek to migrate to a different framework that offers better support for new features, enabling them to stay up-to-date with emerging technologies and best practices. Migration may also be necessary when the adopted DL framework lacks interoperability with modern tools or platforms, which hinders seamless integration. For instance, Hugging Face recently announced the deprecation of TensorFlow[1] and Flax[2] support in their *transformers* library [15]. This decision will require users to transition to supported frameworks to maintain compatibility and leverage Hugging Face's new features and updates. Such ecosystem changes highlight the dependency risks organizations face when committing to a specific framework, especially as library support and feature availability can shift over time. Furthermore, documentation and community support often play a

---

[1] https://www.tensorflow.org/
[2] https://flax.readthedocs.io/en/stable/

key role in migration decisions, as a more active community can offer better resources and timely assistance with issue resolution.

Despite the benefits of migration, there is a lack of approaches designed for NN code. In the literature, some techniques [3, 38, 40] have been proposed to convert trained ready-to-use NNs across frameworks. However, code migration remains largely overlooked, which hinders NN development and the flexibility to adopt new features while reusing existing code. In the absence of adapted techniques, migrating to new DL frameworks generally involves manually rewriting NN code from scratch, a process that typically requires a considerable amount of time, effort, and resources.

In this paper, we contribute an approach to support NN code migration across DL frameworks. Our method builds on a recent work that proposed a Domain Specific Language (DSL) for NNs based on Model-Driven Engineering (MDE) [13]. Specifically, we use the NN metamodel as a pivot to migrate NN code across frameworks. We also develop code generators to automatically produce NN code in the target framework. Moreover, we report on the challenges encountered during the design of the migration approach and how these challenges were addressed. While our migration approach is generic, we focus in this work on the two popular DL frameworks: PyTorch[3] and TensorFlow. Experimental evaluation confirms the success of our approach in migrating NN implementations and shows that the migrated NNs remain functionally equivalent to the originals. We also discuss how the migrated networks can be integrated with the other components of smart systems (e.g., UI, database, ...), which are often modelled using MDE. Our artefacts are made publicly available[4].

The rest of the paper is structured as follows: In Section 2, we discuss the background concepts relevant to our research. In Section 3, we present our migration approach. We describe the challenges encountered during the migration and how they were addressed in Section 4. We discuss how our approach can be integrated into complete system models and evaluate our approach in Section 5 and Section 6, respectively. Finally, we present the related work in Section 7 and conclude this paper in Section 8.

## 2 Background

In this section, we review key concepts necessary to understand our approach, namely neural networks and the BESSER NN metamodel, which are presented in Section 2.1 and Section 2.2, respectively.

### 2.1 Neural Networks

NNs consist of a sequence of layers that are connected together to analyse and process input data. They can model complex relationships, which makes them suitable for tasks, including classification, clustering, regression, and new data generation [17, 23, 28, 43].

Layers are considered the basic building block of NNs. Various types of layers exist, each tailored to extract specific features from the input data to achieve the desired prediction. For instance, convolutional layers are specialised in processing spatial data to extract patterns relevant for the prediction [31]. Recurrent layers are designed for sequential and temporal data, enabling them to capture dependencies within the data sequence [54]. Activation functions

can be applied to the output of layers to introduce non-linearity and help the model capture complex relationships in the data. Besides layers, NNs can be composed of tensorOps that apply low-level operations on the input data, such as addition, multiplication and permutation [30]. Deep networks can also define and call other sub-NNs (sub-neural networks) as part of the main network.

When NN layers are arranged sequentially (i.e., each layer's output is fed as input to the following layer), a **Sequential** architecture can be used to define the network. While this type of architecture is straightforward, it has limitations, as it does not support architectures where layers can receive input from previous non-adjacent layers. For more advanced NNs, where a layer's output is passed into multiple subsequent layers, a **Subclassing** architecture is more suitable, as it provides greater control over layer organization and enables custom forward passes. In many popular frameworks, such as TensorFlow and PyTorch, this is done by defining a class that inherits from a base class where layers are initialized in the constructor and the logic for processing input data is implemented in a dedicated method that defines the forward pass.

After the NN architecture is defined, a dataset is required for training, allowing the NN to learn from examples and adjust its parameters to improve its prediction decision. A loss function is used to guide the learning process, as it calculates the difference between predicted and actual outputs (i.e., the loss), which helps the optimisation process ensure that the network predictions are progressively improved to better match the actual outputs. The training process is further controlled by hyperparameters that determine how the NN learns from the data to optimise the performance, including the learning rate, batch size, and number of epochs. Algorithms such as stochastic gradient descent [6] or Adam [32] can be used to guide the optimisation process, which iteratively updates the network parameters based on the loss value. A test dataset is also required at the end of the training to assess the performance of the trained NN in correctly predicting new, unseen samples.

### 2.2 BESSER NN Metamodel

Our migration approach builds on a recent work that proposes a DSL for neural networks [13]. Specifically, a metamodel has been created to represent NNs, including architectural components such as layers, tensorOps, training and evaluation datasets, and training parameters. A textual notation has also been developed to facilitate NN modelling.

The metamodel and textual notation have been designed as part of BESSER [1], which is a low-code platform aiming to facilitate software development. BESSER proposes a variety of submodules written in B-UML, which represents BESSER's Universal Modelling Language that is inspired by UML [25]. For instance, it includes a structural metamodel that uses concepts from class diagram to create domain models. The NN metamodel was proposed as a submodule of BESSER to enhance its capabilities in supporting smart software development. It is available in the DSL paper [13] and, in the rest of this work, is referred to as *BESSER-NN*.

## 3 Approach

In this section, we present our migration method and describe the different steps involved in the process. The method consists of three

---

[3]https://pytorch.org/
[4]https://github.com/BESSER-PEARL/BESSER-NN-Migration

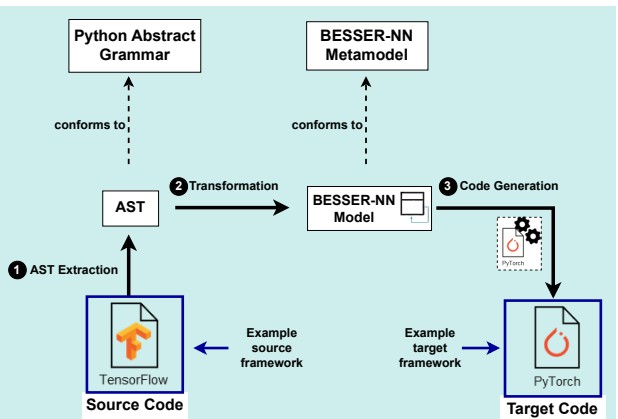

**Figure 1: Overview of our migration approach. TensorFlow is used as the source library and PyTorch as the target library. Migration from PyTorch to TensorFlow is also supported. The approach involves three main steps: 1) AST Extraction, 2) Transformation, and 3) Code Generation.**

main steps: AST Extraction, Transformation and Code Generation. An overview of our approach illustrating the migration steps from TensorFlow code to PyTorch code is presented in Figure 1. In the following, we provide a detailed description of each step to illustrate how they contribute to the migration process.

### 3.1 Source Code AST Extraction

The first step in the migration process is to extract the Abstract Syntax Tree (AST) of the neural network code written in the source framework. AST is a data structure that represents the parsed code as a tree of nodes. Each node in the AST represents a language construct such as functions and variables. In the neural network, AST nodes represent NN components such as layers and tensorOps.

Extracting the AST facilitates source code transformation by providing a representation that preserves logical relationships between its components. AST can be considered a model of the source code as it conforms to Python Abstract Grammar. Consequently, ModelToModel transformations can be applied on the AST as part of the migration process.

We rely on Python's *ast* library[5] to parse the source code since both PyTorch and TensorFlow are primarily used with Python. Our AST code extractor supports both *Sequential* and *Subclassing* NN architectures of the two frameworks. We provide in Figure 2 an illustration of the AST extracted from a *Dense* layer of a neural network [58] written in TensorFlow using the *Subclassing* architecture. The Figure shows that the layer name, its type and parameters are all captured in the AST, providing a structured representation that can be used for the transformation.

### 3.2 Transformation

After extracting the AST, we perform Model-to-Model transformations to obtain the *BESSER-NN* representation of the source code. The *BESSER-NN* model is platform independent and conforms to the

---

[5] https://docs.python.org/3/library/ast.html

*BESSER-NN* metamodel, which is inspired by UML. We use *BESSER-NN* as a pivot model during the migration in order to represent NN code in a format that is independent of the source framework.

We iterate through the AST nodes and map each NN component in the source code AST to its equivalent representation in *BESSER-NN*. Specifically, we map layers, tensorOps and sub-NNs along with their attributes to *BESSER-NN*. We also map the NN hyperparameters, and its training and evaluation constructs, to their equivalents in the pivot *BESSER-NN* model to ensure a faithful transformation.

Since we want our migration to support both *Sequential* and *Subclassing* NN architectures for PyTorch and TensorFlow, we developed four transformation modules, each tailored to a specific combination, namely 1) PyTorch *Subclassing*, 2) PyTorch *Sequential*, 3) TensorFlow *Subclassing* and 4) TensorFlow *Sequential*. We note that transforming TensorFlow code to *BESSER-NN* was challenging as it required additional work to symbolically trace the NN source code and extract layer attributes that could not be retrieved directly from the AST. We provide more details about this process in Section 4.1. At the end of this step, source code AST model is transformed to *BESSER-NN* pivot model.

Figure 2 illustrates how the AST representation of code is transformed into the *BESSER-NN* model. The layer type *Dense* captured in the AST is transformed into the *Linear* type in *BESSER-NN*, along with its attributes: *units* and *activation*, which are mapped to *out_features* and *actv_func*, respectively. We note that the layer attribute *in_features* is not directly available in the TensorFlow source code and is instead obtained by symbolically tracing the code, as explained in Section 4.1.

### 3.3 Code Generation

In this step, the goal is to obtain the final migrated code in the target framework. We develop code generators for PyTorch and TensorFlow so that our approach can migrate code between the two frameworks. We note that our approach also allows for the easy addition of code generators for other frameworks.

The code generation process, which takes the *BESSER-NN* pivot model as input, is divided into two stages: a preprocessing step to refine and prepare the model, followed by Model-to-Text transformations (implemented by the code generators) to map NN concepts to code in the target framework. We opted for Jinja[6] as template engine to develop the code generators, due to its ease of use and flexibility [12]. However, given that Jinja is not optimised for heavy data processing, we relied on Python to preprocess the *BESSER-NN* model (Cf. Section 3.3.1) before generating the code with Jinja (Cf. Section 3.3.2).

*3.3.1 BESSER-NN Model Preprocessing.* We used Python to iterate through the modules classes (i.e., layers, tensorOps, or sub-NNs), collect their attributes, and process this data before passing it to the code generators. We opted for this method to simplify the generation and process some cross references/dependencies between layers. We store the processed information in a dictionary with keys representing the names of the modules defined in the NN. The corresponding values hold details of the modules (e.g., module definition, input tensor variable, etc.).

---

[6] https://jinja.palletsprojects.com/

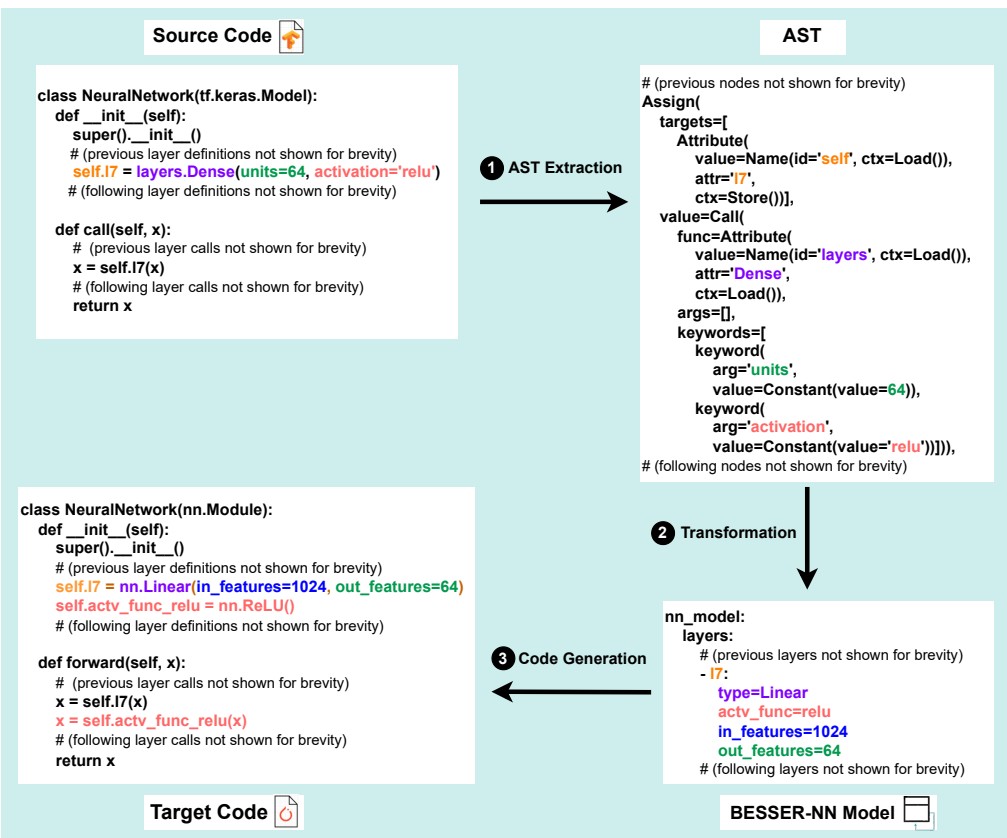

**Figure 2: An illustration of the migration process from TensorFlow to PyTorch, showing the three main steps: 1) AST Extraction, 2) Transformation, and 3) Code Generation. The example focuses on a *Dense* layer of an NN defined using *Subclassing* architecture and shows how the layer name, type, and attributes are mapped to their PyTorch equivalents via BESSER-NN.**

*3.3.2 Code Generators.* Once the preprocessing of the *BESSER-NN* model is complete, it is passed to the code generators that are based on Jinja templates. These generators can be developed exclusively for the *Subclassing* architecture, as it is capable of representing both *Subclassing* and *Sequential* source NNs. However, this method would not allow defining an NN within another class, which is a useful design choice for structuring complex software projects. Consequently, we developed four code generators (i.e., PyTorch (*Subclassing* and *Sequential*) and TensorFlow (*Subclassing* and *Sequential*)) so that our approach can migrate NN code between the four NN architecture type combinations across both frameworks.

We provide in Figure 2 an illustration of the code generation step, showing PyTorch code generated based on the *BESSER-NN* model. The layer name, type, and attributes captured in the *BESSER-NN* are mapped to their PyTorch equivalents. The activation function, presented in the *BESSER-NN* as a layer attribute following the TensorFlow convention, is captured in the generated code as a standalone layer following the PyTorch convention.

## 4 Challenges

During the design of our approach, we faced several challenges that required us to adapt our solution to ensure a successful migration.

In this section, we describe these challenges and how they were addressed in our design to improve its robustness and make sure it can be adopted in a richer variety of transformations beyond mappings among very close frameworks.

## 4.1 Missing Attributes

As our approach supports PyTorch and TensorFlow, the aim is to generate NN code in the target framework that is functionally equivalent to the code in the source framework. To achieve this, we map each layer in the source code to a layer that performs similar operations in the target framework. Similarly, attributes need to be translated to ensure consistent behaviour. We remind that our approach uses a pivot model and all mappings are performed in two stages: first to the pivot model, and then from the pivot to the target framework. This process ensures that key information is collected in the pivot model before it is transferred to the target framework.

To migrate NN code from TensorFlow to PyTorch, the mapping of some layers was challenging. Specifically, there exist attributes that are not defined in TensorFlow layers (i.e., they are inferred automatically from the input data) but are required in PyTorch. These attributes represent the input shape of layers and are denoted in *BESSER-NN* language as: *in_channels* (for *Conv1D*, *Conv2D*

and *Conv3D* layers), *input_size* (for *SimpleRNNLayer*, *LSTMLayer* and *GRULayer*) and *in_features* (for *LinearLayer*). Basically, when migrating TensorFlow code, these attributes are not collected in *BESSER-NN* model as that part of the mapping only captures the attributes defined in the source code layers. When transforming the *BESSER-NN* model to PyTorch, this process generates NN code that conforms to PyTorch syntax. However, the missing input shape attributes cause the code to fail when executed.

To solve this issue, we add an additional step in the transformation process of TensorFlow code. Specifically, we trace the neural network symbolically to collect the required attribute values. By propagating symbolic input tensors through the network, we infer the input shapes of its layers. This enables the generation of PyTorch code that preserves architectural integrity by defining layers along with their required attributes.

## 4.2 Dynamic Assignment of Activation Function

In PyTorch, activation functions implemented in *torch.nn*[7] can be treated as layers in a neural network. Like any other layer, they can be defined in the __*init*__ method and then called in the the *forward* method to modify the output of preceding layers. In TensorFlow, activation functions follow a different terminology: they are passed as parameters to layers, typically by providing a string that represents the activation function's name when defining the layer. *BESSER-NN* follows TensorFlow convention by passing the activation function as a parameter to the layer.

Migrating TensorFlow code implies parsing the activation function name from the layer definition. This supposes that the name of the activation function is explicit in the TensorFlow code so that it can be mapped to a known PyTorch layer. In practice, activation function is not always directly specified as a known string in the layer definition. Instead, it may be assigned to a variable beforehand and referenced indirectly when defining the layer. This can occur when the activation function name is first stored in a variable and later passed to the layer. Another common scenario is when the NN, which follows a *Sequential* design architecture, is defined within a class. In this case, some of the network's attributes, including the activation function, might be set in the __*init*__ method of the class before being used to construct the network.

The use of a dynamically assigned activation function would cause the migration to fail, as the PyTorch code generator expects an explicit string representing a known activation function name, which it can then map to the equivalent PyTorch layer. To overcome this issue, we have designed our PyTorch code generator such that it verifies the value passed to the activation function before generating the NN code. In the case the string value represents a known activation function, the code for PyTorch layer representing this activation function is generated. Otherwise, it is replaced with a call to a custom function that we developed to replace the activation function layer call and dynamically invoke the corresponding PyTorch activation layer at runtime.

We note that no dynamic execution is required at this stage; the name of the activation function is resolved only when the NN is subsequently utilized after the migration. This technique makes our

migration flexible to handle the different ways of defining activation functions in TensorFlow.

## 4.3 CNN Input Compatibility

Convolutional layers apply kernels to extract features from input data and capture spatial relationships. In TensorFlow, input data is received following the channel-last convention, meaning that the input data has the channels as the last dimension. For example, for 2D data, the format would be [batch_size, height, width, channels]. PyTorch, however, receives input data that follows the channel-first convention. For 2D data, the expected format is [batch_size, channels, height, width]. We note that this difference in channel convention applies also to 1D and 3D data.

As the input data flows through the network, its shape changes depending on its initial format and the layers it passes through. Notably, in neural networks with convolutional layers, the intermediate input/output shape might differ between TensorFlow and PyTorch because the two frameworks use different channel ordering conventions. When migrating NN code, the layers can typically be mapped easily between TensorFlow and PyTorch, but their input and output shapes may differ due to the channel ordering conventions of the two frameworks. This would result in inconsistencies/errors in the migrated code since the input/output shape of a layer in the source code might not necessarily correspond to the correct input/output shape of the same layer in the target code.

To obtain consistent code in the target framework, we leverage the *permute*[8] tensorOp, which reorders the dimensions of a tensor according to a specified sequence, in order to make PyTorch follow the TensorFlow channel ordering convention. Specifically, when convolutional layers are present in the PyTorch NN, we permute their input data to align with PyTorch convention of channel-first (given that the data is received in channel-last, following TensorFlow convention). This is because we expect NNs in both frameworks to receive the same input data, and we adopt the TensorFlow convention (i.e., channel-last) as the standard format. After data passes through PyTorch convolutional layers, we permute it back to align again with the TensorFlow convention. This method ensures that the same input data can be processed by neural networks in both frameworks without the need to adapt/change its dimension to conform with each of the two frameworks. Moreover, layers can be defined with the same attributes, since the input shape is always consistent with one convention (i.e., TensorFlow). We note that it is also possible to set TensorFlow to follow the PyTorch convention.

## 5 Integration with the Overall System

Integration is an important aspect to consider during migration to ensure that the model of the NN can be properly merged with the model of the rest of the software system, potentially also generated as part of a migration process (e.g., using frameworks like MoDisco [7]). To achieve this, the NN model must be merged with the software model of the overall system. In this section, we provide an overview of how *BESSER-NN* can be integrated with a more general modelling language, such as UML, to specify NNs as behaviour implementations within a software system.

---

[7]https://docs.pytorch.org/docs/stable/nn.html

[8]https://docs.pytorch.org/docs/stable/generated/torch.permute.html

In practice, the invocation and execution of a neural network can be tied to the declaration of a behaviour for a specific component or entity within a system. For instance, a method executed in one of the states of a state machine could implement a neural network to perform prediction tasks required for decision-making.

Figure 3 shows the integration of *BESSER-NN* with the concepts from the UML modelling language. Integration is done at the meta-model level by combining *BESSER-NN* metamodel and the UML metamodel as we describe in what follows.

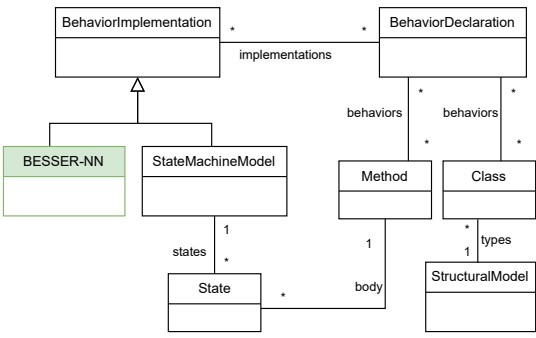

**Figure 3: Integration of *BESSER-NN* with the UML language at metamodel level. This class diagram shows only the relevant classes to illustrate the integration.**

UML enables the specification of various model types, such as state machine models (*StateMachineModel* class in Figure 3) and structural models (*StructuralModel*). Some concepts defined in these models can also define a behaviour declaration (*BehaviorDeclaration*) to describe dynamic aspects of the system. For instance, *Class*es in a *StructuralModel* or *Method*s in a *StateMachineModel* can declare one or more *behaviours*.

On the other hand, the *BehaviorImplementation* concept provides the concrete implementation of behaviours defined by *BehaviorDeclaration*. This implementation can represent specific actions that a class performs under certain conditions or in response to particular events. Both *BESSER-NN* and *StateMachineModel* can be modelled as concrete implementations of a behaviour.

With this extension, any system modelled with UML (including UML-based models produced by model-driven migration pipelines) can define the behaviour of its subsystems using an NN defined with *BESSER-NN*. Integration with other general languages like SysML [19] would be similar.

## 6 Evaluation

We evaluate our approach by considering two main research questions:

- **RQ1:** How effective is our approach in successfully migrating neural network implementations?
- **RQ2:** To what extent does the migration method preserve the functional behaviour of the source neural network?

To answer these research questions, we conduct a series of experiments using five NNs, that we present in Section 6.1. Section 6.2

addresses RQ1, which examines the effectiveness of the migration process. Section 6.3 addresses RQ2, which evaluates whether the functional behaviour of the networks is preserved after migration.

### 6.1 Experimental NNs

We select five NNs from the literature to assess the effectiveness of our approach:

- AlexNet [34] is a convolutional neural network that contains a total of 15 layers.
- LSTM [61] is a recurrent neural network composed of 6 layers.
- VGG16 [56] is a convolutional neural network containing 25 layers.
- CNN-RNN [60] contains a total of 11 layers. It combines both convolutional and recurrent layers and adopts a non-sequential architecture.
- TF-Tutorial [58] is a convolutional neural network introduced in a TensorFlow tutorial. The number of layers in this network is 8.

Overall, these NNs cover a total of 12 distinct layer types and a mix of *Subclassing* and *Sequential* architectures.

### 6.2 RQ1: Migration Success

In this section, we assess the success of the migration and the extent to which it produces structurally correct NN models. To that end, we first migrate the five NNs presented in Section 6.1, and then assess whether the migrated architectures execute without errors. We note that we migrate PyTorch implementation for AlexNet and VGG16, and TensorFlow implementation for LSTM, CNN-RNN and TF-Tutorial. Then, we migrate back to the source implementation. In short, we cover all the possible migration scenarios that we summarize in Table 1. For a given network, a total of eight migration scenarios are considered across the two frameworks, covering both *Subclassing* and *Sequential* architectures.

**Table 1: Migration scenarios from PyTorch to TensorFlow and vice versa**

| | TensorFlow | |
|---|---|---|
| PyTorch | *Subclassing ↔ Subclassing* | *Subclassing ↔ Sequential* |
| | *Sequential ↔ Subclassing* | *Sequential ↔ Sequential* |

After conducting the experiments, our results show that the five NNs can be migrated successfully using our approach. Specifically, AlexNet, VGG16, and TF-Tutorial can be fully migrated across the eight scenarios presented in Table 1. As known by the community, CNN-RNN contains a non-sequential architecture and LSTM can only be defined using a *Subclassing* architecture as it processes an intermediate output before passing it to the next layer. Both architectural designs have limited our migration evaluation only to *Subclassing ↔ Subclassing* across the two frameworks, for which the migration was successfully handled.

Besides migrating code, we also assess whether the produced implementations execute successfully. This step is important to ensure that the migrated NNs do not have structural errors resulting from the migration. We feed dummy input data to the migrated

NNs and examine their behaviour during execution. Our results show that all migrated NNs run successfully on the dummy data.

> **RQ1 answer:** Our migration method effectively converts NN implementations between PyTorch and TensorFlow supporting both *Sequential* and *Subclassing* architectures. The converted NNs executed on dummy inputs, demonstrating the success of the migration.

## 6.3 RQ2: NN Functional Preservation

In this section, we investigate whether the NN code produced by the migration is functionally equivalent to the original. This is important to validate that the approach is accurate and reliable. We consider two evaluation scenarios: first, we feed random inputs to networks initialized with the same weights to compare their outputs, as described in Section 6.3.1; second, we train the networks on benchmark datasets and compare their performance as discussed in Section 6.3.2.

*6.3.1 Validation using Random Inputs.* We conduct our evaluation by (1)- feeding the same input data to the source and target NN and (2)- ensuring that both NNs share the exact same set of weights.

To that end, we begin by randomly initializing one NN and then copy its weights to the migrated NN. Next, we generate random input data to pass it through the two NNs in order to compare their outputs. We repeat this process 100 times for the five experimental NNs and for all the migration scenarios presented in Table 1 to ensure a fair comparison. Since the experimental NNs have different output shapes, we calculate the maximum absolute difference (MAD) between the outputs of the original and migrated networks. For example, AlexNet has 1000 output neurons in its output layer; first, we calculate the absolute difference between each output neuron in the source AlexNet and the neuron at the same position in the migrated AlexNet, then report the largest difference as the MAD. Figure 4 shows the MAD values for migrating the experimental NNs from TensorFlow to PyTorch. The MAD figure for migrating from PyTorch to TensorFlow is similar and is available in our repository.

We observe that all the MAD values are indeed very small—up to $5.96 \times 10^{-8}$—confirming that the original and migrated networks produce similar outputs when fed with the same input. AlexNet, VGG16, and TF-Tutorial, which don't use an activation function in their output layer, show smaller variation, as their outputs directly reflect raw numerical differences between the source and migrated networks. CNN-RNN shows more spread because small differences between the logits of the original and migrated NNs lead to different predicted probabilities, due to the effect of the sigmoid activation used in the output layer. The LSTM network uses the softmax activation function and shows near-zero MADs because small differences in logits between the original and migrated NNs do not affect the output probabilities. We note that previous studies have also reported that, under the same setting, NNs trained with TensorFlow and PyTorch yield slightly different accuracy results [26, 46].

*6.3.2 Validation using Benchmark Datasets.* To further confirm the equivalence of the source and migrated networks, we perform additional experiments to compare their outputs after training.

**Table 2: Accuracy of source and migrated NNs on benchmark datasets using PyTorch and TensorFlow**

| Dataset | Model | TF Accuracy | PyTorch Accuracy |
|---|---|---|---|
| CIFAR-10 | AlexNet | 0.701 | 0.704 |
| CIFAR-10 | VGG16 | 0.753 | 0.775 |
| CIFAR-10 | TF-Tutorial | 0.710 | 0.704 |
| IMDB | CNN-RNN | 0.846 | 0.851 |
| IMDB | LSTM | 0.839 | 0.839 |
| SVHN | AlexNet | 0.933 | 0.941 |
| SVHN | VGG16 | 0.927 | 0.932 |
| SVHN | TF-Tutorial | 0.895 | 0.894 |
| SST-2 | CNN-RNN | 0.811 | 0.825 |
| SST-2 | LSTM | 0.817 | 0.796 |

Specifically, we train each pair of source and migrated networks on two benchmark datasets to thoroughly examine the equivalence of their behaviours. AlexNet, VGG16, and TF-Tutorial are trained and evaluated on the CIFAR-10 [33] and SVHN [44] datasets, which contain image samples. For LSTM and CNN-RNN networks, the IMDB [41] and SST-2 [57] datasets are used. All pairs are trained under the same settings for 10 epochs to examine their behaviour and compare their performance. The number of epochs was inspired by the TensorFlow tutorial from which our TF-Tutorial model is taken, as it was sufficient for most of the NNs to reach a reasonable prediction accuracy. For the LSTM and CNN-RNN models trained on the SST-2 dataset, we extended the training to 50 epochs, as 10 epochs were not sufficient for the models to converge on this dataset. After training, we evaluate the performance of all source and migrated pairs of NNs and report their prediction accuracy in Table 2.

By comparing the accuracy scores of the source and migrated NNs, we find that they achieve similar prediction performance, with the largest difference of 0.022 observed for VGG16 on the CIFAR-10. The small differences are consistent with known numerical differences that arise when training NNs independently across different frameworks [9, 26, 62]. The results show that, across all datasets, the migrated NNs yield prediction results consistent with those of the original NNs.

> **RQ2 answer:** The migration method effectively preserves the functional behaviour of the original neural network, as confirmed by experiments with both random inputs and benchmark datasets.

## 7 Related Work

In this section, we review prior approaches that are closely related to our work. In particular, we discuss the research area of neural network interoperability and the approaches proposed in that context, as presented in Section 7.1. Methods for converting neural networks to a different deep learning framework are reviewed in Section 7.2.

### 7.1 Interoperability of NNs across Frameworks

Interoperability across frameworks has been a significant challenge when developing neural networks. In the literature, some

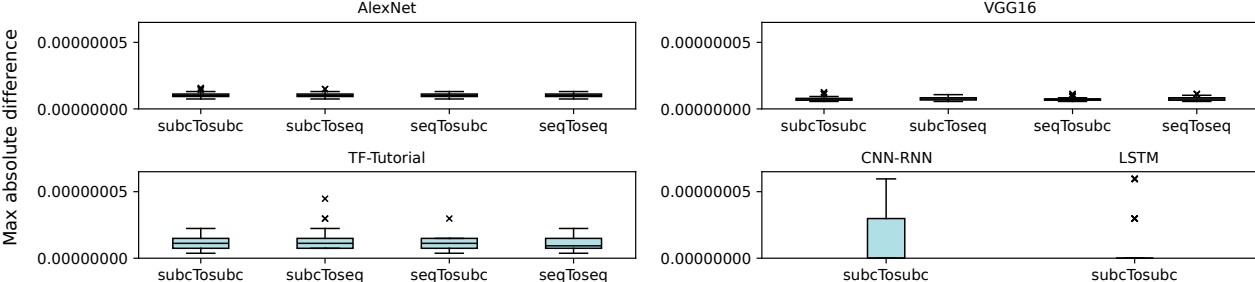

**Figure 4: Max absolute differences (MAD) for NNs migrated from TensorFlow to PyTorch. XToY denotes migration from architecture X in the source NN to Y in the target NN. 'subc' and 'seq' are used to refer to *Subclassing* and *Sequential*, respectively.**

approaches have been proposed to enable the integration of different frameworks within the same script.

Ivy [37] is a library that enables interoperability between ML frameworks allowing NNs to run seamlessly across PyTorch, TensorFlow, JAX, and NumPy. EagerPy [53] is a framework that provides a unified tensor interface, enabling users to write tensor manipulation code that executes seamlessly on PyTorch, TensorFlow, JAX, and NumPy. For instance, a norm function defined with EagerPy can take as input tensors defined in any of these frameworks without compatibility issues. While these frameworks enable cross-platform compatibility, existing code still needs to be adapted or rewritten to their syntax to achieve interoperability across platforms.

Unlike the above approaches, ours does not aim to make NN frameworks interoperable in the sense of enabling simultaneous cross-framework compatibility. Instead, it focuses on converting NN **code** from a source to a target framework where the target code is native to the target framework, as if it was originally written in that framework. This enables leveraging all the features of the target framework.

### 7.2 Conversion of NNs across Frameworks

Several approaches have been proposed to convert NNs across frameworks. ADELT [21] is a source-to-source transpiler that relies on an LLM for code skeleton transpilation and a learned dictionary for API keyword substitution. While ADELT performs source-code-level migration, the use of an LLM inherently introduces non-determinism, and keyword substitution alone cannot handle structural differences such as one-to-many API mappings, which ADELT acknowledges as a failure case. Our approach addresses such cases natively through an explicit rule-based pivot model that also covers the full training pipeline.

Another line of work focuses on converting trained NNs from one framework to another. CoreML [2, 3] is a framework developed by Apple to convert AI models into a unified representation, facilitating their integration into applications. MMdnn [38] transforms trained NNs into intermediate computation graphs before converting them to the desired framework. Nobuco [40] targets NNs trained with PyTorch to transform them into Keras/TensorFlow. pytorch2keras [52] converts trained CNNs from PyTorch to Keras. caffe-tensorflow [14] converts Caffe NNs into TensorFlow, though its authors noted that not all NNs can be converted due to framework mismatches.

Some approaches rely on a framework-agnostic format for representing trained NNs before conversion. Open Neural Network Exchange (ONNX) [49] is a format for representing NNs developed in a variety of frameworks including PyTorch and TensorFlow. It relies on a computational graph, built-in operations and data types to ensure a proper NN representation. ONNX derived libraries [16, 42, 47, 48, 51, 63] convert AI models between ONNX format and other standard machine learning frameworks. To transform a model to ONNX format, these tools take a model object as input rather than source code, and therefore operate at the level of a computation graph extracted from a model instance. Consequently, training configurations such as loss functions and optimizers, which are not part of the model object, are not captured or migrated. Similarly, Neural Network Exchange Format (NNEF) [24] facilitates the transfer of NNs from the original framework into various inference engines, operating on trained NN models rather than source code. Prior works in AI compilers, such as TVM [11], MLIR [35], and DyCL [10], have also explored intermediate representations for cross-framework model transformations. These approaches target hardware-level inference optimization through low-level tensor operations, whereas BESSER-NN captures NN architectural structure and training constructs to perform source code migration.

Our approach differs from these works in that it converts NN **code** from a source to a target framework. The produced code can be used to train the network in the target framework. The above model converters, however, do not involve any code migration. Only the final trained NN, ready for deployment, is converted to the target framework. Although migrating trained NNs may be sufficient for inference-only scenarios, this method limits opportunities to inspect, modify, or retrain the model in the new framework. By migrating the NN definition itself (rather than a fixed, trained model), our approach better supports iterative experimentation, adaptation, and retraining, making it particularly suited for scenarios where the developer needs to inspect or adapt the NN code in the target framework. Deployment-oriented frameworks such as TensorFlow Lite [22], which optimize trained models for on-device inference, are therefore not targeted by our approach. Furthermore, our approach integrates the migrated NN into broader system models at the UML/SysML metamodel level (Cf. Section 5), enabling the NN to be embedded as a behavioural component in model-driven software pipelines, which is outside the scope of the above conversion tools. Moreover, there have been concerns about inconsistencies,

behavioural differences, and performance degradation between the source and converted models produced by these NN conversion tools [29, 50]. A failure analysis of ONNX model converters reveals that converters are responsible for the majority of conversion defects in PyTorch and TensorFlow [29]. Further work [36] confirms that existing model conversion solutions suffer from unreliable performance, proposing a systematic verification method to ensure interoperability between original and converted models. Our validation (Cf. Section 6) shows that our approach avoids some of these issues by preserving the functional behaviour of NNs after migration.

## 8 Conclusion

This work proposes a migration approach for converting neural network code across frameworks. At its core, our method relies on a pivot NN specification as an intermediate and more abstract representation of NNs to facilitate the transformation among any pair of source and target NN libraries. We also discuss the challenges to migration and how they were overcome in our approach. Integration with the overall software system is also an important aspect that we address in our paper. We validate the effectiveness of our approach using five NNs migrated across two popular deep learning frameworks: PyTorch and TensorFlow. Our experiments, which involved training and evaluating the source and migrated NNs on four benchmark datasets, demonstrate that the migration produces NN code that is functionally equivalent to the original implementations.

As future work, we plan to extend our migration approach to support other AI frameworks. We would also like to expand *BESSER-NN* to incorporate more advanced concepts from the deep learning and, in general, AI field. For instance, with the addition of concepts such as transformers, graph neural networks, reinforcement learning and even wrapping interactions with LLMs via the modelling of the model context protocol[9]. This would enable the migration of more sophisticated smart software systems. Moreover, we plan to devise specific refactoring transformations to help neural network designers to have a better NN structure based on best practices from the field. This would be implemented as an internal transformation where the source and target library would be the same. Furthermore, we plan to develop a method to migrate NN testing code, ensuring that all code related to the neural network is properly translated to the target framework.

## Acknowledgments

This project is supported by the Luxembourg National Research Fund (FNR) PEARL program, grant agreement 16544475.

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
