# OpenReview forum: "Towards Migrating Neural Network Implementations"
_ACM.org/AIWare/2026/Conference — AIware 2026_

### Official Review · Reviewer_4o5p · 2026-02-26

**Rating:** 3
**Confidence:** 4

**Review:**

Overall, the paper is well-written and easy to follow. In addition, the authors provide a code repository for the system they propose.

However, I have some considerable concerns regarding this work.

**Novelty:**
The primary contribution of this work is a system for converting deep learning models across frameworks at the code level. However, prior research has already explored this problem. In particular, "ADELT: Transpilation Between Deep Learning Frameworks" [1] presents a transpilation system that performs source level conversion across frameworks such as PyTorch and Keras. The overall objective of automated framework-to-framework translation at the code level is therefore closely aligned with the goal of this contribution. Although the technical mechanisms differ, the problem formulation and intended outcome substantially overlap.

That said, the abstraction mechanism proposed in this submission, specifically the introduction of an intermediate model representation prior to regeneration, has the potential to more clearly distinguish the work. Such an abstraction layer could offer advantages in semantic preservation and structural reasoning compared to purely syntactic transpilation approaches. However, as authors state, this mechanism is also attributed to prior work, and therefore presents limited novelty by itself.

I recommend that the authors explicitly articulate how their abstraction fundamentally differs from prior transpilation systems such as ADELT, provide a structured comparison such as a conceptual pipeline diagram contrasting their method with ADELT and similar approaches, and clarify whether their design enables capabilities that code-level transpilers cannot support, for example semantic validation, structural optimization, or portability guarantees.

**Related Work:**
The related work section in this submission includes important conversion tools such as MMdnn and widely-used model converters. However it contains room for improvement, as it omits to cite and briefly present findings of work associated with model conversions. For example, [2] conducts failure analysis for deep learning framework model converters. While this work utilizes ONNX and does not focus in code-oriented conversions and transpilation, it clearly presents a considerable set of challenges. It would be beneficial if the authors consider the findings of such works and compare how they apply in their migration process. As another example, [3] presents challenges associated with the interoperability across model conversions, which could also be compared across the findings of this contribution.

**Limited Evaluation:**
The evaluation focuses only on PyTorch and TensorFlow. The absence of experiments involving constrained or deployment-oriented frameworks (e.g., TensorFlow Lite) which might have restrictions in features and operations (and, therefore, present additional challenges towards model conversions) raises considerable concerns about scalability and generalizability. In addition, it would be useful to explore how effective the proposed methodology is when converting models of increased complexity (e.g., small-sized LLMs or more complex multi-branch architectures).

**Interoperability:**
In the introduction, the authors claim that interoperability is a major factor for converting models across deep learning frameworks, stating that "migration may also be necessary when the adopted DL framework lacks interoperability with modern tools or platforms, which hinders seamless integration". However, in the related work section, they state that “unlike the above approaches, ours does not aim to make NN frameworks interoperable. Instead, it focuses on converting NN code from a source to a target framework where the target code
is native to the target framework, as if it was originally written in that framework”, which effectively downplays the role of interoperability. This inconsistency weakens the paper’s argument, as it overlooks one of the two most critical elements in model conversion.

Overall, the paper is clearly presented but presents limited novelty in relation to existing transpilation literature, with limited experimental breadth and insufficient positioning within prior migration research. I recommend that the authors emphasize potential points of novelty, demonstrate methodology generalizability by utilizing a more comprehensive set of experiments and conduct a more comprehensive literature review, clearly comparing existing works with this contribution.

[1] https://dl.acm.org/doi/10.24963/ijcai.2024/694

[2] https://dl.acm.org/doi/10.1145/3650212.3680374

[3] https://reference-global.com/article/10.2478/jaiscr-2023-0016

**Important update: The authors provided adequate clarifications to my questions. Given that they adequately address all aspects as they mention, conduct adequate comparison to related work and provide details on scalability, generalizability, as well as potential future work directions, the paper can be accepted.**

**Summary:**

The paper introduces a framework for migrating Deep Neural Networks (DNNs) across different deep learning frameworks. The authors perform model conversions between PyTorch and TensorFlow using a metamodel-based transformation, emphasizing code-level rather than graph-level conversions. They evaluate their approach on five different neural networks, performing conversions between PyTorch and TensorFlow, and discuss the challenges encountered during this process.

---

> ### Author Response · Authors · 2026-03-17
> **Responses to Reviewer 4o5p Comments**
>
> We thank the reviewer for the detailed and constructive feedback. We address each concern below.
>
> **W1: Novelty**
> We acknowledge that ADELT is a relevant related work that was overlooked and will be added to Section 7. While both approaches aim to automate the migration of deep learning code across frameworks, ADELT is a data-driven approach that learns mappings from a corpus, whereas our approach is fully rule-based and relies on an explicit structured intermediate model of the NN.
> ADELT relies on an LLM for code skeleton transpilation and a learned dictionary for API keyword substitution. The use of an LLM inherently introduces non-determinism into the process. Our approach, by contrast, transforms the source code into an explicit pivot model (BESSER-NN) that captures layers, tensorOps, hyperparameters, training constructs, and architectural topology in a framework-independent format, from which target code is generated through defined mappings.
> This pivot model also allows our approach to handle cases that fall outside the scope of keyword substitution, such as attributes implicitly inferred in TensorFlow but required explicitly in PyTorch, or activation functions assigned dynamically to a variable. Importantly, ADELT itself acknowledges that one-to-many API mappings, such as a TensorFlow layer with an inline activation function that must become two separate constructs in PyTorch, represent a failure case that neither ADELT nor Codex can solve without additional synthetic training data. Our approach handles this natively by design, as illustrated in Figure 2 of the paper.
> Finally, our approach covers the full training and evaluation pipeline, not only the model architecture, and enables the migrated NN to be embedded as a component within broader software systems modelled in standard languages such as UML or SysML. As described in Section 5, the NN can be composed with other system components such as databases, state machines, or user interfaces within a model-driven development pipeline, which is outside the scope of transpilation approaches such as ADELT.
>
> **W2: Related Work**
> We thank the reviewer for pointing out these missing references. Jajal et al. is already cited in Section 7.2 of the paper but will be discussed more explicitly: their findings on node conversion failures and semantically incorrect models in ONNX converters directly relate to the challenges documented in Section 4 of our paper, particularly the missing attributes and channel ordering issues that arise when mapping between TensorFlow and PyTorch. Lee et al. will be added to Section 7, with a discussion of how the interoperability challenges they identify in trained model conversion complement the source-code migration challenges we address.
>
> **W3: Limited Evaluation**
> Regarding deployment-oriented frameworks such as TensorFlow Lite, our approach targets source code migration for development, retraining, and integration purposes rather than deployment optimisation. TensorFlow Lite operates on trained models optimised for on-device inference and does not support the full training pipeline that our approach is designed to migrate, which would require a fundamentally different problem formulation. We will clarify this scope distinction more explicitly in Sections 1 and Section 7.
> Regarding more complex architectures, we agree this is a valid direction for future work. As stated in Section 8, extending support to transformers, graph neural networks, and LLM interaction patterns is already planned as a next step.
>
> **W4: Interoperability**
> We acknowledge that the term "interoperability" appears in two distinct contexts in the paper. In the introduction, it refers to the compatibility of a framework with external tools and platforms: when a framework loses such compatibility, migration to a supported framework becomes necessary to maintain integration with the broader ecosystem. In Section 7.1, it refers to cross-framework runtime compatibility, as provided by tools such as Ivy and EagerPy, which allow code to execute across multiple frameworks simultaneously. Our approach addresses the first context: it produces native target-framework code, enabling full access to the target framework's tools, features, and ecosystem. Achieving runtime interoperability, as offered by Ivy or EagerPy, is a separate and orthogonal goal that our approach does not pursue. We will make this distinction explicit in the revised text.
>
> We believe these clarifications address the reviewer's core concerns and we welcome any follow-up questions.

---

> ### Comment · Reviewer_4o5p · 2026-03-18
>
> Thank you for the clarifications, these are important and in depth. I am willing to update my review, given that the authors address all the comments aforementioned. More importantly, I would like to see a clear comparison and elaboration of similarities and differences with related work such as ADELT, the including references, as well as other potential related work on interoperability, model conversions, etc. You will also need to clearly define the meanings of interoperability, as well as address concerns related to scalability, and mention directions to related work. Overall, I think the work is valuable, but should be better supported in all the aspects aforementioned, which is *vital* for its acceptance.

---

### Official Review · Reviewer_ZYdX · 2026-03-08

**Rating:** 4
**Confidence:** 3

**Review:**

Strengths

1. The paper goes beyond previous research on transforming pre-trained models and instead focuses on migration at the source code level. This is more useful in practice because the migrated results remain editable, trainable, and can be integrated back into larger software processes. Additionally, adopting a transformation design from an abstract syntax tree to a pivot model is a reasonable structured transformation approach, rather than treating migration as a black-box transformation problem.

2. The evaluation is also fairly convincing for a first step. The migrated networks execute successfully, and the reported results on datasets such as IMDB and SVHN suggest that the migrated implementations preserve strong performance in practice.

Weaknesses

1. The paper motivates well why source-code migration is different from trained-model conversion, but I still miss a more direct comparison with existing conversion approaches. In particular, it would be useful to understand the trade-offs in success rate, coverage, and practical effort more clearly.

**Summary:**

This paper studies migration of neural network implementations across deep learning frameworks. Instead of converting trained models only, it tries to migrate the NN code itself. The proposed pipeline extracts the AST from source code, maps it to a framework-independent pivot model based on BESSER-NN, and then generates code in the target framework. The current implementation focuses on TensorFlow and PyTorch, and the experiments on five networks suggest that the migrated code can run successfully and largely preserve the original behaviour.

---

> ### Author Response · Authors · 2026-03-17
> **Responses to Reviewer ZYdX Comments**
>
> We thank the reviewer for the positive assessment and the constructive feedback.
>
> **W1: Missing Direct Comparison with Existing Conversion Approaches**
> We appreciate this remark and acknowledge that a more explicit comparison would strengthen the paper. Section 7.2 already discusses existing conversion tools such as ONNX, tf2onnx, MMdnn, Nobuco, and pytorch2keras, and positions our approach with respect to them. The main difficulty in conducting a direct quantitative comparison is that these tools operate on a fundamentally different input: they take trained models with frozen weights and produce a runtime-ready artefact for inference. Our approach takes source code as input and produces native, editable, trainable code in the target framework. The two therefore address different stages of the development lifecycle and different user needs, which makes a direct comparison on metrics such as success rate or coverage difficult to set up.
>
> We agree that a more explicit discussion of when each type of approach is applicable, and what each requires from the user in terms of effort and prerequisites, would be a useful addition. We will expand Section 7.2 accordingly.

---

### Official Review · Reviewer_1K5K · 2026-03-11

**Rating:** 3
**Confidence:** 5

**Review:**

## 2. Strengths

### 1. Clear Problem Motivation

The paper addresses a real and practical software engineering problem: migrating neural network implementations across different frameworks. This issue frequently arises in industrial environments where machine learning frameworks evolve rapidly and models need to be ported between ecosystems.

### 2. Potential for Automation

Automating the migration process can significantly reduce engineering effort and minimize human errors that often occur during manual rewriting of neural network models.

### 3. Clear Presentation

The paper is well-written, and the overall presentation is clear and easy to follow.

---

## 3. Weaknesses

### 1. Limited Evaluation Scope

The evaluation appears limited in scale, involving only five model architectures and two frameworks. The paper would benefit from a more comprehensive evaluation, including:

1. A wider variety of model architectures,
2. Additional deep learning frameworks, and
3. Larger and more complex models (e.g., Transformers or large language models) to better demonstrate the robustness and generalizability of the proposed approach.

### 2. Missing Comparison with Existing Interoperability Solutions

A major limitation of the paper is the absence of comparisons with existing **model interoperability standards**, particularly ONNX. Considering that ONNX was specifically designed to provide a framework-agnostic representation for deep neural networks, the novelty of the proposed pivot model is unclear without such a comparison.

In particular, the paper does not evaluate the proposed pivot model against ONNX, which is a widely adopted standard for cross-framework neural network representation. ONNX already provides:

1. A framework-agnostic intermediate representation,
2. Conversion tools for frameworks such as PyTorch, TensorFlow, and others, and
3. A mature ecosystem supporting model portability and deployment.

Therefore, the paper should clearly discuss:

1. **How the proposed pivot model differs from ONNX**
2. **Why a new intermediate representation is necessary**
3. **What specific limitations of ONNX the proposed approach aims to address**

Without such a discussion or comparison, it is difficult to understand the advantages and novelty of the proposed method.

### 3. Lack of In-Depth Discussion on Model Inconsistency

Based on the results reported in Table 2, there appear to be inconsistencies between the original models and the migrated models. It would be valuable for the authors to provide a deeper analysis of the causes of these inconsistencies. For example, they could investigate whether the discrepancies arise from floating-point numerical differences [1, 2], implementation differences across frameworks, or limitations of the migration tool itself.

### 4. Missing Related Work on Neural Network Abstraction

Some existing work on neural network abstraction and intermediate representations is missing. For example, prior research in AI compilers and deep learning systems [3, 4, 5] has explored intermediate representations and cross-framework model transformations. Discussing these works would help better position the proposed approach within the broader research landscape.





[1] Your Compiler is Backdooring Your Model: Understanding and Exploiting Compilation Inconsistency Vulnerabilities in Deep Learning Compilers.


[2] Understanding and Mitigating Numerical Sources of Nondeterminism in LLM Inference

[3] DyCL: Dynamic Neural Network Compilation Via Program Rewriting and Graph Optimization

[4] TVM: An Automated End-to-End Optimizing Compiler for Deep Learning

[5] MLIR: A Compiler Infrastructure for the End of Moore's Law

**Summary:**

## Summary

This paper addresses the challenge of migrating neural network (NN) implementations across different deep learning frameworks. As machine learning frameworks rapidly evolve, organizations may need to migrate models from one framework to another due to performance improvements, ecosystem changes, or new requirements. However, such migration is often difficult and labor-intensive because of differences in APIs, supported layers, and implementation paradigms.

To tackle this issue, the authors propose an automated migration approach based on an intermediate pivot neural network model. The core idea is to abstract a neural network implementation into a framework-independent representation and then generate equivalent code for a target framework. This architecture allows the migration logic to be modular and extensible. Instead of writing direct translators between every pair of frameworks, only two mappings are needed for each framework:framework → pivot and pivot → framework.

---

> ### Author Response · Authors · 2026-03-17
> **Responses to Reviewer 1K5K Comments**
>
> We sincerely thank the reviewer for the thorough and constructive feedback. We address each weakness below.
>
> **W1: Limited Evaluation Scope**
> We acknowledge that extending the evaluation to more architectures and frameworks would strengthen the paper. When selecting the five architectures, we aimed to cover as much structural diversity as possible: the evaluation covers 12 distinct layer types, includes both Sequential and Subclassing architectural styles, covers models ranging from 6 to 25 layers, and includes convolutional, recurrent, and hybrid CNN-RNN models with a non-sequential architecture ensuring that the migration is evaluated across structurally heterogeneous configurations.
> Supporting advanced architectures such as transformers and LLMs, as well as additional deep learning frameworks, is planned as future work, as stated in Section 8. Regarding transformers and LLMs, we plan to expand BESSER-NN to incorporate the relevant deep learning concepts, which would enable the migration of more sophisticated architectures. Regarding additional frameworks, the modular design of our approach (where only two mappings are required per new framework) makes such extensions straightforward, and we plan to target frameworks such as JAX/Flax in future work.
>
> **W2: Missing Comparison with Existing Interoperability Solutions**
> We appreciate this concern, but we respectfully point out to our comparison with ONNX in Section 7.2 of the paper, where we explain the key distinction between our approach and ONNX-based conversion tools. The fundamental difference is as follows:
> • ONNX (and related tools such as tf2onnx, MMdnn, Nobuco, pytorch2keras) operate on trained, ready-to-deploy models. They convert the final weights and computation graph for inference in a target runtime. The resulting artefact is not source code that can be inspected, modified, or retrained in the target framework.
> • Our approach migrates NN source code, producing native target-framework code as if it had been originally written in that framework. This enables iterative experimentation, retraining, debugging, and integration into model-driven pipelines.
> Furthermore, the literature has documented concerns about inconsistencies, behavioural differences, and performance degradation in ONNX-based conversions [24, 41 in the paper]. Our validation (Section 6) shows that our approach avoids several of these issues by preserving functional equivalence at the code level.
> We will nonetheless strengthen Section 7.2 to make this comparison more explicit and prominent, as we acknowledge that the distinction may not be sufficiently emphasized in the current version.
>
> **W3: Lack of In-Depth Discussion on Model Inconsistency**
> We thank the reviewer for this comment. The small differences observed in Table 2 between source and migrated models are not introduced by the migration process itself, but from well-documented framework-level implementation details such as floating-point arithmetic ordering, default weight initialization, and numerical precision during independent training in TensorFlow and PyTorch [9, 21, 38]. The reviewer's cited works on compiler-induced numerical inconsistencies and non-determinism in deep learning further support this interpretation.
> The fidelity of the migration is assessed through the MAD analysis reported in Figure 4, where source and migrated models are loaded with identical weights and fed the same inputs, isolating the effect of migration from any training variability. The resulting differences are on the order of 1e-8, confirming that the migration process itself does not introduce numerical inconsistencies. We will add clarification in Section 6.3 to make this distinction more explicit and will reference the reviewer's suggested works as additional supporting evidence.
>
> **W4: Missing Related Work on Neural Network Abstraction**
> We thank the reviewer for this observation. Works such as TVM , MLIR , and DyCL operate at the level of compiler intermediate representations (IRs) and address cross-platform optimization for model inference and execution.
> Our pivot model (BESSER-NN) is conceptually different in scope and purpose:
> • AI compiler IRs (TVM, MLIR) represent computational graphs and low-level tensor operations optimized for hardware execution.
> • BESSER-NN represents NNs at a higher abstraction level, capturing the architectural structure (layers, connectivity, hyperparameters) in a framework-independent metamodel aligned with Model-Driven Engineering (MDE) principles.
> Our goal is source code migration for development and retraining, not hardware-level inference optimization. The two lines of work are therefore complementary rather than competing. We will add a dedicated paragraph in Section 7 situating our approach with respect to AI compiler IRs, citing TVM, MLIR, and DyCL, and clarifying this distinction.
>
> We believe these clarifications address the reviewer's core concerns and we welcome any follow-up questions.

---

> > ### Comment · Reviewer_1K5K · 2026-03-19
> > **Response to Authors**
> >
> > ## Missing Comparison with Existing Interoperability Solutions
> >
> > Authors claim that ONNX operate on trained, ready-to-deploy models, which is not actually True, ONNXRuntime runs on ready-to-deploy models, but ONNX is a IR to abstract model. and there are many tools can convert Pytorch, Tensorflow model to ONNX and convert ONNX format back to  Pytorch, Tensorflow. Given the existing tools, it is necessary to show the advantage of the proposed tool.
> >
> >
> > https://github.com/ENOT-AutoDL/onnx2torch
> > https://github.com/gmalivenko/onnx2keras
> > https://github.com/onnx/keras-onnx
> > https://docs.pytorch.org/docs/stable/onnx.html

---

> ### Author Response · Authors · 2026-03-21
> **Responses to Reviewer 1K5K Comment**
>
> We thank the reviewer for this clarification and apologize for the imprecision in our previous rebuttal response. When we used the term "ONNX" in our rebuttal, we were referring to ONNX-based conversion tools rather than the ONNX format itself. The paper already describes ONNX correctly as "a format for representing NNs … ", and we stand by that description.
> Regarding the reviewer's point that it is necessary to show the advantage of the proposed tool, we highlight the following concrete distinctions:
>
> **Source code migration.** Our approach takes Python source code as input, parses it into an AST, and produces fully native Python source code in the target framework. The generated code is written exactly as a developer would have originally written it in that framework, which can be directly inspected, modified, and extended. The tools the reviewer cites (onnx2torch, onnx2keras, keras-onnx, torch.onnx) all take a model object as input, not source code. For example, torch.onnx.export explicitly requires a model instance and sample input tensors (https://docs.pytorch.org/docs/stable/onnx.html). This is the fundamental reason BESSER-NN is necessary: ONNX operates at the level of a computation graph extracted from a model instance, whereas our approach operates at the level of source code structure, capturing constructs that simply do not exist in a computation graph.
>
> **Training pipeline migration.** Because ONNX-based tools operate on a model instance, they capture only the forward pass of the network. Training configurations such as optimizers, loss functions, and hyperparameters are not part of the model object and are therefore not captured. Our transformation step explicitly maps layers, architecture, hyperparameters, training configurations, and evaluation constructs to the BESSER-NN pivot model, migrating the full development pipeline and not only the inference structure.
>
> **MDE integration.** Our approach integrates the migrated NN into broader system models at the UML/SysML metamodel level (Section 5), enabling the NN to be embedded as a behavioural component in model-driven software pipelines. This is outside the scope of ONNX and its ecosystem tools entirely.
>
> **Documented limitations of ONNX-based conversion.** Two empirical studies already cited in our paper document conversion failures and report inconsistencies, behavioural differences, and performance degradation between source and converted models produced by ONNX-based tools. Our MAD evaluation confirms functional equivalence at the order of 1e-8, a level of precision that ONNX-based conversions have been shown not to always preserve.
>
> We will revise Section 7.2 to make these distinctions more explicit and precise and we thank the reviewer for this thorough and constructive feedback.